# Rapid transmission and tight bottlenecks constrain the evolution of highly transmissible SARS-CoV-2 variants

Emily E. Bendall [1], Amy P. Callear[2], Amy Getz[2], Kendra Goforth[2], Drew Edwards[2], Arnold S. Monto[2], Emily T. Martin[2] & Adam S. Lauring [1,3] ✉

Transmission bottlenecks limit the spread of novel mutations and reduce the efficiency of selection along a transmission chain. While increased force of infection, receptor binding, or immune evasion may influence bottleneck size, the relationship between transmissibility and the transmission bottleneck is unclear. Here we compare the transmission bottleneck of non-VOC SARS-CoV-2 lineages to those of Alpha, Delta, and Omicron. We sequenced viruses from 168 individuals in 65 households. Most virus populations had 0–1 single nucleotide variants (iSNV). From 64 transmission pairs with detectable iSNV, we identify a per clade bottleneck of 1 (95% CI 1–1) for Alpha, Delta, and Omicron and 2 (95% CI 2–2) for non-VOC. These tight bottlenecks reflect the low diversity at the time of transmission, which may be more pronounced in rapidly transmissible variants. Tight bottlenecks will limit the development of highly mutated VOC in transmission chains, adding to the evidence that selection over prolonged infections may drive their evolution.

Viral populations are often subject to multiple bottleneck events as they evolve within and between hosts. These bottlenecks drastically reduce the size and genetic diversity of the population, which will affect how new mutations spread through host populations[1,2]. In the setting of a tight transmission bottleneck, most mutations that arise within a host are not propagated between them. Bottlenecks also reduce the virus's effective population size, which captures the number of virions that reproduce and genetically contribute to the next generation; selection is less effective in smaller populations. Therefore, tight bottlenecks constrain adaptive evolution by limiting the spread of newly arising mutations and reducing the efficiency of selection on these mutations along transmission chains. Many viruses, such as HIV[3,4], influenza[5], and SARS-CoV-2[6–10], have tight bottlenecks, with 1–3 distinct viral genomes transmitted.

The size of the transmission bottleneck may be impacted by viral dynamics, route of infection, or molecular interactions at the virus-host interface. For example, it has been suggested that transmissibility, or force of infection, may influence bottleneck size. Increased transmissibility may lead to wider bottlenecks in several ways. First, increasing the infectious dose, perhaps through increased shedding in the donor host or increased intensity of contact, can lead to wider bottlenecks as shown in experimental infections of influenza A virus[11,12] and tobacco etch virus[13]. Additionally, the number of virions that initially infect cells is directly related to bottleneck size[14]. More transmissible viruses may have an increased ability to infect individual cells, such as through increased receptor affinity or escape from intrinsic or innate immunity.

While early studies of SARS-CoV-2 transmission estimated a tight transmission bottleneck, the last 20 months of the pandemic have witnessed the emergence of highly transmissible variants of concern (VOC). In December 2020, B.1.1.7 (Alpha) was detected for the first time with a substantial increase in transmissibility over previous SARS-CoV-2 lineages[15]. Since then, additional variants of concern characterized by an increase in transmissibility have arisen. The Alpha, Beta, Gamma, Delta, and Omicron VOC are 25–100% more transmissible than the original Wuhan strain[16]. There are multiple and overlapping mechanisms for the increased transmissibility in SARS-CoV-2 that may influence bottleneck size, including increased binding to ACE2[17–20],

[1]Department of Microbiology and Immunology, University of Michigan, Ann Arbor, MI, USA. [2]Department of Epidemiology, University of Michigan, Ann Arbor, MI, USA. [3]Division of Infectious Diseases, Department of Internal Medicine, University of Michigan, Ann Arbor, MI, USA. ✉e-mail: alauring@med.umich.edu

increased viral shedding[21,22], innate immune evasion[23], rapid cellular penetration[18], and alternative entry pathways[24,25].

Here we explore the relationship between viral transmissibility and transmission bottlenecks by comparing bottleneck size across multiple VOC and pre-VOC lineages. We sampled viral populations from two household cohorts in Michigan, obtaining high depth of coverage sequence from 168 individuals in 65 households. We found that bottleneck size did not vary significantly between transmission pairs infected with pre-VOC lineages and those infected with highly transmissible Alpha, Delta, or Omicron (BA.1) lineages. This tight bottleneck estimate was driven by the limited diversity in the donor host at the time of transmission.

## Results

We used high depth of coverage sequencing to characterize SARS-CoV-2 populations collected from individuals enrolled in a prospective surveillance cohort (HIVE) and a case-ascertained household cohort (MHome). There were 65 multiply infected households (infections ≤14 days apart) with 168 cases. COVID-19 severity was relatively mild, with only one individual requiring hospitalization. High quality, whole genome sequences (see Methods) were obtained with technical replicates from 131 cases. Depth of coverage was generally high and iSNV frequency was similar across both replicates (Supplementary Fig. 1). There were five households that had consensus sequences inconsistent with household transmission (Supplementary Fig. 2). Of these five, two households with two individuals each were excluded. In two households, there was a single individual whose consensus sequence differed from the others and was excluded. In the final household, the consensus sequences were consistent with two separate transmission pairs, and these were analyzed separately. All 5 households with multiple introductions were due to either Delta or Omicron viruses, consistent with high community prevalence during these waves[26]. The final transmission analysis dataset included 45 households, 110 individuals, and 134 possible transmission pairs (Table 1). Alpha (B.1.1.7), Gamma (P.1), Delta (AY.3, AY.4, AY.39, AY.44, AY.100), and Omicron (BA.1, BA.1.1) were represented in these households. Variants of interest included one household with Lambda (C.37).

### Transmission dynamics

There was rapid transmission of SARS-CoV-2 in the sampled households. The median serial interval ranged between 2 and 3.5 with no significant difference observed between clades (df = 4, F = .879, $p = 0.483$, Fig. 1a, Supplementary Figs. 3 and 4). Households with Delta and Omicron had a greater range of serial intervals. Viral specimens were collected soon after symptom onset in both household studies, with clade-specific medians ranging from 2–6.5 days. Omicron had a shorter time between index symptom onset and sample collection for sequencing than non-VOC (df = 3, F = 8.138, $p < 0.001$) and Alpha ($p = 0.01$) (Fig. 1b, Supplementary Figs. 3 and 4). This is likely due to the number of Omicron cases in HIVE households, which had a shorter time between index symptom onset and sample collection for sequencing than MHome households (df = 1, F = 15.363, $p < 0.001$).

### Within-host viral diversity

We further examined the timing of index case sampling by plotting RT-qPCR Ct values for all index case specimens. In nearly all cases, the index cases were sampled at or near peak viral shedding (Fig. 1c). Therefore, our sequence data for the index cases should be reflective of the genetic diversity present in donor hosts when risk of household transmission was highest. Consistent with the short time between the infection onset and sample collection, we found low genetic diversity in nearly all specimens (Fig. 2a). A majority (56/110, 51%) had no iSNV above the 2% frequency threshold; 42% (46/110) of samples had 1–2 iSNV; and 7% (8/110) had ≥ 3 iSNV. There were no specimens with more than 5 iSNV. Fifty-two percent of iSNV were present at <10% frequency within hosts, Fig. 2b.

### Estimated transmission bottlenecks

Bottleneck size is calculated based on shared diversity between members of a transmission pair. Within each household, possible transmission pairs included the index case as donor and each household contact as a recipient, and household contacts as donors for other household contact recipients. While the majority of sampled households had only two cases, 12 had three cases, and 4 had four cases (Fig. 3a). The number of possible transmission pairs per household ranged from 1 to 12 (Supplementary Data 1). When we compared the frequency of iSNV in the donors and recipients, we found only a single shared iSNV—C29708T (noncoding)—in 6 possible transmission pairs from a single household (Fig. 3b). This iSNV was present in all three individuals in the household at a frequency of 0.56, 0.97, and 0.24 respectively. All other iSNV were either absent (frequency of 0) or completely fixed (frequency of 1) in the other individual of the transmission pair for all households. This pattern is highly suggestive of a narrow bottleneck.

We used the beta binomial model[27] to obtain a quantitative estimate of the transmission bottleneck for individual transmission pairs and by clade. Because bottleneck size can only be calculated when there are iSNV in the transmission donor (see Fig. 2a), we were able to use 64 potential pairs in this analysis (Supplementary Data 1). All VOC clades had an overall bottleneck size of 1 (Alpha, Delta, Omicron: 95% CI 1:1, Gamma: 95% CI 1:7). The Non-VOC clades had an overall bottleneck size of 2 (95% CI 2:2), which was driven entirely by the single shared iSNV in one household. The 6 transmission pairs in this household exhibited bottlenecks of 2, 4, and 6 (Supplementary Data 2). All other transmission pairs had a bottleneck size of 1 inclusive of all clades. Across all transmission pairs, the upper bound of the 95% confidence interval varied greatly, from 1 to 200, the maximum bottleneck size we evaluated (Supplementary Data 2).

We were stringent in our variant calling criteria and required iSNV to be present in both sequencing replicates, because false positive iSNV can artifactually inflate bottleneck estimates[7,28–30]. To ensure that our stringency did not lead to an underestimate, we re-analyzed our dataset after merging sequencing reads across the technical replicates. This had only a small effect on the number of iSNV identified in each specimen (Supplementary Fig. 5). Thirty-nine out of 110 specimens still had no iSNV present, and all but 2 specimens had ≤8 iSNV. The remaining two specimens had 25 and 57 iSNV. The newly detected iSNV

## Table 1 | Sample size by clade for transmission analyses

|                                                            | Non-VOC | Alpha | Gamma | Delta | Omicron | Total |
|------------------------------------------------------------|---------|-------|-------|-------|---------|-------|
| Individuals with successful sequencing                     | 22      | 21    | 3     | 25    | 40      | 111   |
| Households with successful sequencing[a]                   | 11      | 7     | 1     | 12    | 15      | 46    |
| Possible transmission pairs                                | 26      | 34    | 2     | 19    | 55      | 134   |
| Transmission pairs included in bottleneck analysis[b]      | 15      | 19    | 1     | 12    | 17      | 64    |

[a]Households that have 2 or more individuals with successful sequencing.
[b]Only includes transmission pairs where there are iSNV in the donor.

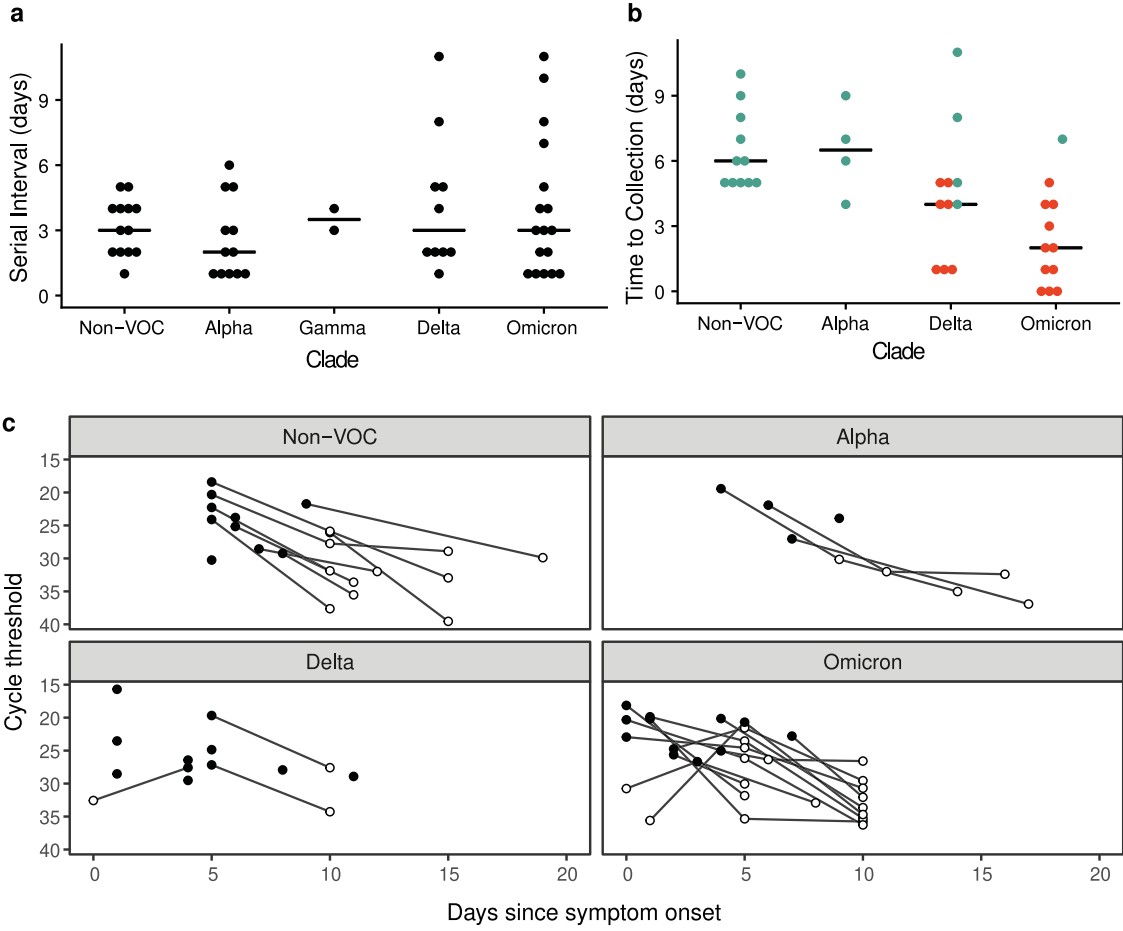

**Fig. 1 | Serial interval and timing of sample collection. a** Days between index symptom onset and household contact symptom onset for the indicated clades. "Non-VOC" includes all lineages not designated as a WHO variant of concern. No Beta variant transmission pairs were analyzed. **b** Days between symptom onset and collection of the sequenced specimen for the index case. Index cases from MHome are indicated in teal, and index cases from HIVE are indicated in red. Omicron had a shorter time between index symptom onset and sample collection for sequencing than non-VOC (ANOVA, df = 3, F = 8.138, $p < 0.001$) and Alpha. HIVE households had a shorter time than MHome households (ANOVA, df = 1, F = 15.363, $p < 0.001$). **c** RT-qPCR cycle threshold values (inverted y-axis) for all specimens collected from index cases. Sequenced specimens are indicated with filled circles.

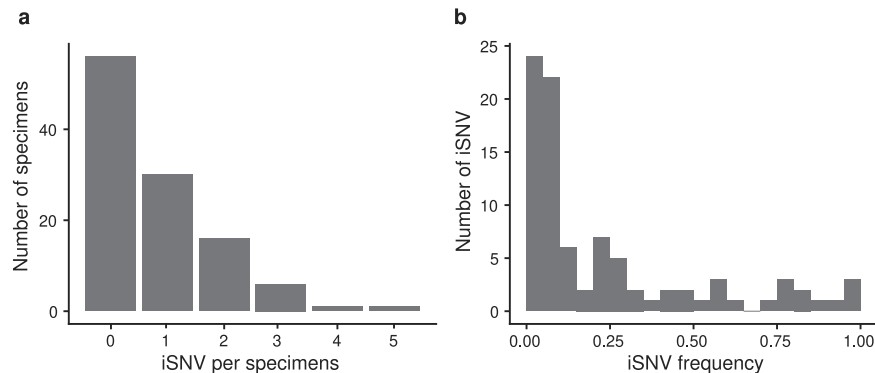

**Fig. 2 | Genetic diversity in sequenced specimens. a** Histogram of the number of iSNV per specimen. **b** iSNV frequency histogram.

in the merged dataset tended to be present at very low frequency (<3%) and shifted the iSNV frequency distribution toward lower values (Supplementary Fig. 5). In this lower stringency dataset, an additional 19 transmission pairs had iSNV in the donor. However, the bottleneck sizes for all clades were identical to the previous estimates (Supplementary Data 3). This suggests that the tight bottlenecks we estimated were not due to overly stringent variant calling.

## Discussion

Here, we used in depth sequencing of two well-sampled household cohorts to define the relationship between transmissibility and transmission bottleneck size. We found that all clades exhibited short serial intervals in our households and low genetic diversity in specimens collected close to the time of transmission. Because of this limited genetic diversity, we estimated a tight bottleneck. In line with

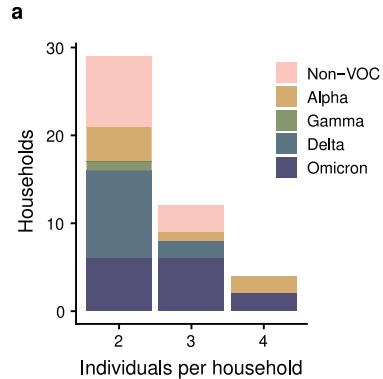

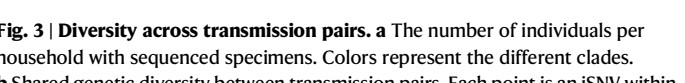

**Fig. 3 | Diversity across transmission pairs. a** The number of individuals per household with sequenced specimens. Colors represent the different clades. **b** Shared genetic diversity between transmission pairs. Each point is an iSNV within a transmission pair. Red points indicate mutation C29708T, which was shared in a single household (see text).

bottleneck estimates for first-wave lineages of SARS-CoV-2 we found that VOC clades had a bottleneck of 1 and non-VOC had a bottleneck of 2. These very tight bottleneck estimates were robust to reductions in the stringency in variant-calling.

Consistent with prior studies of SARS-CoV-2 and other viruses, we found low genetic diversity within and between hosts. Allowing for slight differences due to analytic pipelines, previous studies have largely reported low within-host genetic diversity in SARS-CoV-2[6,9,31–34]. Much of this diversity is not shared between hosts, and therefore, multiple studies in different settings have measured a tight transmission bottleneck for SARS-CoV-2[6–10]. Tight bottlenecks appear to be broadly applicable across routes of infection and viral family. Potato Y virus (0.5–3.2) and Cucumber mosaic virus (1–2), both transmitted by aphids[35,36], along with Influenza (1–2), HIV[3,4], Venezuelan equine encephalitis[37], and HCV[38] have tight bottlenecks.

Additionally, we demonstrate that increased transmissibility, whether through force of infection or immune escape, doesn't change the bottleneck size for SARS-CoV-2. Low genetic diversity can constrain transmission bottleneck estimates. If only a single genotype is transmitted, a bottleneck of 1 is inferred. However, multiple virions of a single genotype can found a population. Transmission of multiple genetically identical virions is more likely when there are few iSNV and/or when iSNV are at a low frequency and when bottlenecks are already reasonably narrow (i.e., <10). Regardless of the ability to detect the actual number of founding virions, the biological effect is the same—no genetic diversity is being transmitted from the donor to the recipient. In our comparison of non-VOC and VOC, the short generation time of SARS-CoV-2 does not allow for diversity to accumulate in the donor, much less transmit.

These effects may be exaggerated in highly transmissible variants if time to transmission is shortened. While we did not find variant-specific differences in serial interval in our cohorts, multiple studies that explicitly modeled generation time during household transmission have shown shorter generation times as the pandemic has progressed. Even before variants of concern arose, the generation time of SARS-CoV-2 was decreasing[39], and this trend continued as variants of concern arose with Delta (3.2 days) exhibiting a shorter generation time than Alpha (4.5 days)[40]. A shortening of generation could potentially have a larger impact on bottleneck size for other viruses, particularly those that generate more diversity than SARS-CoV-2 prior to transmission.

Our work highlights how transmission bottlenecks, as typically measured, are distinct from infectious dose. Within-host processes in the recipient influence bottleneck size, because not all virions that initiate an infection go on to establish a genetic lineage[1]. After infection begins, stochastic loss (genetic drift) during exponential growth, superinfection exclusion, cell-to-cell heterogeneity, and host immune response cause some virions to be lost[41]. These within-host processes combined with the starting genetic diversity cause bottleneck size to, in many cases, be smaller than the infectious dose. In experimental systems, genetic barcoding and more frequent sampling of donor and recipient hosts can be used to link bottlenecks to infectious dose and identify lineages that are lost[12,42].

Our study is subject to at least three limitations. First, in all studies of natural transmission, there is always some ambiguity about who infected whom. In two-infection households, it is possible that both were exposed to a common donor outside the household, and in households with >2 cases, there are multiple possible transfection pairs. Because individuals who don't transmit to each other are unlikely to share diversity, incorrect pairing will underestimate the bottleneck[5]. However, we found that all transmission pairs had equal bottlenecks even when we tested mutually exclusive transmission pairs. Second, virus populations may be spatially segregated within hosts, and the transmitted population may not have been well sampled by our analysis of nasal swabs[43–47]. However, given the low viral diversity identified in nearly all cases, even spatially segregated viral populations are likely to be genetically similar to each other. Third, rare diversity may have been under sampled in the donors and recipients due to the sensitivity of our sequencing approach, including missing iSNV at sites below our coverage threshold (<400x). This possibility was addressed in our analysis of merged technical replicates. Given that more common variants (10–50% frequency) were not shared between hosts, it is unlikely that even perfect detection would find shared iSNV at lower frequencies.

Understanding how different viral properties promote or impede evolution is critical for predicting and effectively monitoring the course of the COVID pandemic. The tight bottlenecks we have estimated for SARS-CoV-2 VOC will both limit the spread of new mutations and reduce the effectiveness of natural selection. Weakened selection will inhibit the evolution of new lineages and may be especially important for new VOC. Whereas other lineages may evolve through non-selective mechanisms, such as genetic drift, the existing VOC have exhibited strong signals of prior positive selection at the time of their emergence[16,48–50]. The tight bottlenecks identified here will limit the development of highly mutated VOC in transmission chains of acutely infected individuals, adding to the evidence that selection over prolonged infections in immunocompromised patients may drive the evolution of SARS-CoV-2 variants of concern[6,15,51,52].

## Methods

### Households and sample collection

Households were enrolled through two household cohorts in Southeast Michigan—MHome and the Household Influenza Vaccine Evaluation Study (HIVE). MHome is a case ascertained household cohort in

which households are recruited following identification of an index case who meets a case definition for COVID-like illness and is positive for SARS-CoV-2 by clinical testing. Households in this study were enrolled between November 18, 2020 and January 19, 2022 with individuals aged <1 to 76. HIVE is a prospective household cohort (individuals aged <1 to 77) with year-round surveillance for symptomatic acute respiratory illness. We identified all HIVE households with ≥1 individuals positive for SARS-CoV-2 between June 1, 2021 and January 18, 2022. For both studies, written informed consent (paper or electronic) was obtained from adults (aged >18). Parents or legal guardians of minor children provided written informed consent on behalf of their children. Participants were compensated for their time and effort. Both study protocols were reviewed and approved by the University of Michigan Institutional Review Board (HIVE: HUM118900 & HUM198212, MHome: HUM180896).

In MHome, index enrollees meeting the case definition (at least one the following: cough, difficulty breathing, or shortness of breath; or at least two of the following: fever, chills, rigors, myalgia, headache, sore throat, new loss of smell or taste) with a positive clinical test result within the last 7 days are invited to enroll themselves and their household members. Nasal swabs were collected on days 0, 5, and 10 after enrollment for all participating household members. For HIVE, study participants were instructed to collect a nasal swab at the onset of illness, with weekly active confirmation of illness status by study staff. Eligible illness was defined as two or more of cough, nasal congestion, sore throat, chills, fever/feverish, body aches, or headache (for participants 3 years & older) or two or more of cough, runny nose/nasal congestion, fever/feverish, fussiness/irritability, decreased appetite, trouble breathing, or fatigue (for participants under 3 years old). If a participant had symptoms of a respiratory illness, specimens were collected from all members of that household on days 0, 5, and 10 of the index illness. For both cohorts all samples were nasal swabs that were self-collected, or in the case of young children, parent-collected following an established protocol[53]. In both cohorts, participants were questioned about the day of symptom onset and duration of symptoms. In MHome, the index case was defined as the individual with the earliest symptom onset date. If two or more individuals shared the earliest onset date, they were considered to be co-index cases.

## Viral sequencing
All samples were tested by quantitative reverse transcriptase polymerase chain reaction (RT-qPCR) with either the TaqPath COVID-19 Combo Kit from Thermofisher (MHome) or CDC Influenza SARS-CoV-2 Multiplex Assay (HIVE). We sequenced the first positive sample in each individual with a cycle threshold (Ct) value ≤30 from each individual. RNA was extracted using the MagMAX viral/pathogen nucleic acid purification kit (ThermoFisher) and a KingFisher Flex instrument. Sequencing libraries were prepared using the NEBNext ARTIC SARS-CoV-2 Library Prep Kit (NEB) and ARTIC V3 (MHome, through November 10, 2021) and V4 (MHome, after November 10, 2021; HIVE) primer sets. After barcoding, libraries were pooled in equal volume. The pooled libraries (up to 96 samples per pool) were size selected by gel extraction and sequenced on an illumina MiSeq (2 × 250, v2 chemistry). We sequenced all samples in duplicate from the RNA extraction step onwards, randomizing sample position on the plate between replicates.

We aligned the sequencing reads to the MN908947.3 reference using BWA-mem v 0.7.15[54]. Primers were trimmed and consensus sequences were generated using iVar v1.2.1[55]. Intrahost single nucleotide variants (iSNV) were identified for each replicate separately using iVar[55] with the following criteria: average genome wide coverage >500x, frequency 0.02–0.98, p-value $<1 \times 10^{-5}$, variant position coverage depth > 400×. We also masked ambiguous and homoplastic sites (Supplementary Data 4)[56]. Finally, to minimize the possibility of false

variants being detected, the variants had to be present in both sequencing replicates. Indels were not evaluated.

## Delineation of transmission chains and SARS-CoV-2 lineages
Alignments of consensus sequences within each household were manually inspected. We considered infections to be consistent with household transmission if the consensus sequences differed by ≤2 mutations[31,57]. We excluded individuals whose consensus sequences were inconsistent with household transmission but retained the rest of the household if there was evidence of household transmission among the other members. Households were split and analyzed separately if the consensus sequences supported multiple independent transmission chains within the household. If necessary, we reassigned the index case, so that the index case was part of the transmission chain.

For households with genetically linked infections, we further analyzed all samples with high quality sequencing (>500× coverage) from households with ≥2 members. We used Nextclade to annotate clades and variants of concern[58]. We used the WHO definition to classify variants of concern (i.e., Alpha, Beta, Gamma, Delta, and Omicron: BA1)[59]. Variants of interest were included in the non-variants of concern group for all analyses.

## Infection dynamics
Serial intervals were calculated as the time between symptom onset of the index and each household contact and compared across clades using an ANOVA. Additionally, the times between symptom onset and sample collection for index cases were calculated. Serial intervals and time to sampling across clades were compared using an ANOVA followed by a Tukey HSD. We also compared the Ct values from the nucleocapsid gene of sequenced samples and the other positive non-sequenced samples for index cases.

## Bottleneck estimation
We defined the possible transmission pairs within each household as follows: the index was allowed to be the donor for household contacts, and the household contacts were allowed to be donors to each other. The only case in which the index case was allowed to be the recipient was when there were co-index cases. Co-index cases were allowed to be both donor and recipient with respect to the other co-index. After defining the transmission pairs, we applied the approximate beta-binomial approach[27]. This method accounts for the variant calling frequency threshold and stochasticity in the recipient after transmission. We estimated the bottleneck size for each transmission pair individually and also calculated an overall bottleneck size for each clade using a weighted sum of loglikelihoods[27]. We re-calculated the above bottleneck estimates after merging replicate aligned fastq files to examine the impact of our variant calling strategy.

## Statistics and reproducibility
No statistical method was used to predetermine sample size. No data were excluded from the analyses, except as described in the Result. The experiments were not randomized. The Investigators were not blinded to allocation during experiments and outcome assessment.

# Data availability
Raw sequencing reads are available on the NCBI short read archive (https://www.ncbi.nlm.nih.gov/sra) under BioProject PRJNA889424. All other data, including source data for Figures, may be found in Supplementary Data 1–4.

# Code availability
Scripts necessary to replicate the analyses are available on github (https://github.com/lauringlab/SARS-CoV-2_VOC_transmission_bottleneck, https://doi.org/10.5281/zenodo.7415147).

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

## Acknowledgements

We thank all individuals who participated in this study. This project has been funded in part with Federal funds from the National Institute of Allergy and Infectious Diseases, National Institutes of Health, Department of Health and Human Services, under Contract No. 75N93021C00015 and R01 AI148371 and from the Centers for Disease Control and Prevention, under U01IP001034.

## Author contributions

Conceptualization, A.S.L.; Formal Analysis, E.E.B.; Investigation, E.E.B. and A.G.; Resources, A.P.C., D.E., K.G., A.S.M., and E.T.M.; Data Curation, E.E.B., A.P.C., A.G., D.E., K.G.; Writing Original Draft, E.E.B. and A.S.L.; Writing Reviewing and Editing, E.E.B., A.S.M., E.T.M., and A.S.L.; Supervision, A.S.L.; Funding Acquisition, A.S.M., E.T.M., and A.S.L.

## Competing interests

The authors declare no competing interests.
