## [Peer Review File · Nature Communications]

REVIEWER COMMENTS

Reviewer #1 (Remarks to the Author):

This is an excellent paper that analyzes the transmission bottleneck across VOCs and non-VOCs in SARS-CoV-2. I found it to be well written and interesting. I did have one concern about how the paper was pitched and how the results are presented. In the abstract and in part of the results/discussion, the authors write that they inferred a bottleneck of 1 in non-VOCs and 2 in VOCs. This is a little misleading. As the authors themselves specify in the results, the bottleneck of 2 is driven by one household. Moreover, examination of supplementary table S2 reveals that the CI values for the inferences are quite wide, often ranging between 1 and 200. A rigorous comparison should thus include comparing the CIs between VOCs and non-VOCs, and my guess would be that this comparison would be non-significant. If so, the abstract should be rewritten to emphasize this.

Some minor comments:

1. It would be useful to zoom in on the lower frequencies in Fig. S1 to see how much concordance there is between the technical replicas in this region.
2. Some genomics regions have quite low coverage, which would impact SNV detection. Could the authors elaborate on how this would affect their calculations
3. Could there be some bias in the way the sample is collected? In other words, the requirement for an SNV present in the donor (which makes sense) implies selection for samples with higher diversity. Could the authors discuss this.

Reviewer #2 (Remarks to the Author):

This is a well-argued and tightly written paper, which presents results of sequence-derived estimates of transmission bottleneck size of SARS-CoV-2. Consistent with previous reports, the authors find that the bottleneck of transmission is generally narrow, with most infections originating from a single viral genome, and a minority originating from two. The full dataset comprises 168 individuals from 65 households, of which 64 individuals from 46 households are included in the final analysis. Importantly for the reliability of results, 131 technical replicates were taken forward from the point of RNA extraction, and results used to validate the intrahost variant (iSNV) calling, as well as to perform sensitivity analyses on the final bottleneck estimates. The findings are largely not novel, but

they corroborate previous reports of narrow bottlenecks in SARS-CoV-2 transmission, substantially expand on previously available sample numbers, and extend this to newer variants of concern (VOCs) including Delta and Omicron.

Major comments:

As the authors indicate, there are substantial differences between the VOCs in timing of sampling, which they attribute to differences between the studies that make up the dataset. This implies that the time to accumulate mutations is different for the different VOCs in this dataset. Some discussion of this is warranted: assuming similar mutation rates for all VOCs, would there be less observable within-host diversity for Omicron simply due to the study design (more rapid collection)? Or, given that almost all donor samples are collected around peak viral load regardless of VOC, does this imply that Omicron infections reach peak VL faster than previous VOCs and should in fact be less diverse within-host? Some context from current literature would be helpful in interpreting this. As far as I'm aware there are no studies claiming that Omicron infections tend to harbour less intrahost diversity overall than pre-Omicron variants, and in fact at least one study that claims the opposite is true (Al-Khatib 2022, iScience).

As the authors acknowledge, the main finding of this study is of necessity constrained by the very limited viral diversity within the donor cases. This represents the situation in most infections at the time of transmission, where viral load is high, but the infection is relatively short-lived and intrahost diversity is correspondingly low. Similar observations have been made by others over the course of the pandemic, including Wang et al 2021 doi: 10.3389/fmed.2021.585358 (not cited by the authors, but certainly relevant), and Lythgoe et al 2021, both of which report similarly narrow transmission bottlenecks, at least in pre-VOC viruses. It is unfortunately very difficult to say to what extent this common pattern holds true when it comes to the rarer, but epidemiologically relevant, cases of transmission from individuals with longer-lasting and more diverse infections, possibly to other individuals susceptible to multiple infection. The authors make a convincing case for narrow bottlenecks once data are constrained to the type of infection where donor diversity is minimal, but the population-level limitations should be expanded on in the discussion: had there have been any persistent (or severe) infections within the dataset, or any known immunocompromised individuals, would the expectation of narrow bottlenecks still hold? Is diversity of the sampled disease presentations something future studies should specifically address?

The paper does not mention severity of presentation, but since these are individuals in community-based studies, my assumption is that most are likely to be cases of relatively mild self-limiting disease. Is it possible to include a description of the range of severity in among the cases, age range, vaccination status if known, treatment, and any notable demographic factors?

I have a few minor comments which appear below.

Abstract, line 31 - please specify how many samples were VOC v non-VOC

Results line 87 / Figure S2 - can you justify the exclusion on the basis of >2 SNPs? In the cases where putative transmission pairs differed at a small number of consensus positions, as in households HH32 and HH33, was there any evidence of within-host diversity at these positions in the two samples?

Results L109 - is "trending" used in the sense of drawing a trend line?

Fig S4 legend - please state what the asterisks indicate.

Reviewer #3 (Remarks to the Author):

The size of the transmission bottleneck of a viral pathogens determines how viral genetic diversity is transmitted between hosts. This is an important quantity to estimate to understand how natural selection and drift will act across chains of transmission and ultimately at the host population scale. This paper uses epidemiologically confirmed transmission pairs to estimate the size of the SARS-CoV-2 bottleneck. While similar studies are often constrained by the size of the available dataset the present study has data from 134 putative transmission pairs. Therefore, the authors are able to ask if viral evolution (i.e. the emergence of VOCs) has led to changes in the bottleneck size over time. This is the first study that I am aware of to evaluate this question and therefore represents an important contribution to the literature. The manuscript is very clearly written with pleasant data visualizations and the conclusions in the paper are largely supported by the data analysis.

I do, however, have some comments which I would prefer to see addressed prior to publication.

Major Comments

-Throughout the manuscript (lines 32, 77-78, 160-161, 177-178, etc) the authors assert that limited genetic diversity resulted in a small bottleneck. First, it is not clear whether the authors are implying that the low genetic diversity results in a small estimated bottleneck (lines 160-161) or a small actual

bottleneck (lines 32, 77-78, 177-178). The distinction is important as it determines whether this is a comment about the limitations of the inference method or a comment on the biological reality. Regardless, however, I do not think I necessarily agree with either claim.

First, I do not think it is true that low genetic diversity in the donor host should bias the inference method downwards. While it is true (as is noted) that when there are no iSNVs in the donor host it becomes impossible to estimate the bottleneck size, if there are any variants for the inference method to act on, then there should be no inherent bias due to low diversity in the donor host. The method is more concerned with the similarities in variant frequencies between donor and recipient hosts. For example, the author's own Figure 3 shows that there are actually quite a few variants present in the donor hosts (all of the black dots with x value > 0) but most of them are present at either 0% or 100% in the recipient host, which is why the estimated bottleneck is low. The same quantity of data would provide very different estimated bottleneck sizes if these donor-recipient frequencies were more similar. While I haven't tested this myself and therefore am willing to be proven wrong, I think that if there was only a single iSNV identified in both the donor and recipient host and if it were present at near identical frequencies in each then the beta binomial method should estimate a large bottleneck size.

Second, I further do not think low diversity in the donor host would indicate that the actual bottleneck size is small. Particularly, the phrasing on lines 177-178 is confusing to me. The beta binomial method estimates the census bottleneck size. That is, the number of individual virions that seeded the sampled diversity in a recipient host. These individual virions do not necessarily have to be genetically distinct. While it would be impossible to infer, a completely homogeneous donor viral population could have a census bottleneck size > 1 even if its genetic bottleneck (number of genetically distinct virions) is necessarily fixed at 1. This is a common misconception in the literature but is fairly unambiguously defined in Sobel Leonard et al. 2017: "This transfer is characterized by a transmission bottleneck, defined as the size of the founding pathogen population in the recipient host."

Minor comments:

-I'm having trouble understanding how it was concluded that in nearly all cases the index cases were sampled at peak viral shedding (Line 110-111) from the analysis of only two data points per person, especially given findings that infectiousness peaks right about at symptom onset (<https://doi.org/10.1038/s41591-020-0869-5>), considerably earlier than most of the samples in this analysis. I do not think this has a significant impact on the results or conclusions, however.

-It's not totally clear which sites are masked as the De Maio analysis has evolved over time. I was able to find it by digging through the GitHub repo but I'd like to see the

"ncov_references/problematic_sites_v7.txt" file included as a supplementary table to aid in tracking that information down.

-This is not something that I think needs to be done, but I would like to see some visualization of the Nb estimates or to see them in a main table purely for readability. It can be frustrating as a reader to open a paper looking for a very specific estimate (in this case the Nb across clades) and not see it presented in any figures or tables in the main text.

-I'll leave this to the discretion of the editor but I prefer to see code moved to a DOI'd repository prior to publication. GitHub repos can be deleted and thus are not a permanent storage solution.

Overall I think this is a wonderful analysis that robustly addresses an important question in the field.

We thank the three reviewers for their thoughtful comments and general enthusiasm for the rigor, scope, and significance of our manuscript. We have addressed all reviewer comments below, with our responses in italics.

Reviewer #1 (Remarks to the Author):

This is an excellent paper that analyzes the transmission bottleneck across VOCs and non-VOCs in SARS-CoV-2. I found it to be well written and interesting. I did have one concern about how the paper was pitched and how the results are presented. In the abstract and in part of the results/discussion, the authors write that they inferred a bottleneck of 1 in non-VOCs and 2 in VOCs. This is a little misleading. As the authors themselves specify in the results, the bottleneck of 2 is driven by one household. Moreover, examination of supplementary table S2 reveals that the CI values for the inferences are quite wide, often ranging between 1 and 200. A rigorous comparison should thus include comparing the CIs between VOCs and non-VOCs, and my guess would be that this comparison would be non-significant. If so, the abstract should be rewritten to emphasize this.

We have now clarified in the abstract that the confidence intervals presented in the original submission are indeed at the clade level. Table S2 shows the individual level confidence level intervals, which are of course wider. Thus, from a strictly statistical sense, the clade level estimates are different. In our original abstract in our preprint (shortened on submission due to length restrictions from the journal), we did highlight how the non-VOC clade-level bottleneck of 2 was driven by a single shared iSNV in a single household. This is highlighted in the results, and we don't think we really "pitched" or made much of the difference between 1 and 2, given how little data supported this statistically significant difference. We could include a sentence on this one household if a longer abstract is allowed.

Some minor comments:

1. It would be useful to zoom in on the lower frequencies in Fig. S1 to see how much concordance there is between the technical replicas in this region.

>>> This is a good suggestion. We have added an inset in a revised Fig. S1.

2. Some genomics regions have quite low coverage, which would impact SNV detection. Could the authors elaborate on how this would affect their calculations

>>> We required coverage of >400x at a position in order to call a variant. On re-review of our coverage data, there are just two amplicons where coverage is ~500x in several samples. In our experience and published work (see, for example, Valesano et al. PLOS Pathogens 2021), coverage of 500x is sufficient to call an iSNV at 2% frequency. We have added a sentence to the limitations section about missed iSNV due to imperfect sensitivity and how that could affect our calculations.

3. Could there be some bias in the way the sample is collected? In other words, the requirement for an SNV present in the donor (which makes sense) implies selection for samples with higher diversity. Could the authors discuss this.

>>> All samples were collected in the same way and all samples with Ct<30 were sequenced. There was no requirement for iSNV in the donor. Approximately half of the samples had no iSNV. We did not include these in the bottleneck analysis, as it is not possible to estimate a bottleneck when there are no iSNV in the index/donor (see comments addressing reviewer 3 below).

Reviewer #2 (Remarks to the Author):

This is a well-argued and tightly written paper, which presents results of sequence-derived estimates of transmission bottleneck size of SARS-CoV-2. Consistent with previous reports, the authors find that the bottleneck of transmission is generally narrow, with most infections originating from a single viral genome, and a minority originating from two. The full dataset comprises 168 individuals from 65 households, of which 64 individuals from 46 households are included in the final analysis. Importantly for the reliability of results, 131 technical replicates were taken forward from the point of RNA extraction, and results used to validate the intrahost variant (iSNV) calling, as well as to perform sensitivity analyses on the final bottleneck estimates. The findings are largely not novel, but they corroborate previous reports of narrow bottlenecks in SARS-CoV-2 transmission, substantially expand on previously available sample numbers, and extend this to newer variants of concern (VOCs) including Delta and Omicron.

Major comments:

As the authors indicate, there are substantial differences between the VOCs in timing of sampling, which they attribute to differences between the studies that make up the dataset. This implies that the time to accumulate

mutations is different for the different VOCs in this dataset. Some discussion of this is warranted: assuming similar mutation rates for all VOCs, would there be less observable within-host diversity for Omicron simply due to the study design (more rapid collection)? Or, given that almost all donor samples are collected around peak viral load regardless of VOC, does this imply that Omicron infections reach peak VL faster than previous VOCs and should in fact be less diverse within-host? Some context from current literature would be helpful in interpreting this. As far as I'm aware there are no studies claiming that Omicron infections tend to harbour less intrahost diversity overall than pre-Omicron variants, and in fact at least one study that claims the opposite is true (Al-Khatib 2022, iScience).

As the authors acknowledge, the main finding of this study is of necessity constrained by the very limited viral diversity within the donor cases. This represents the situation in most infections at the time of transmission, where viral load is high, but the infection is relatively short-lived and intrahost diversity is correspondingly low. Similar observations have been made by others over the course of the pandemic, including Wang et al 2021 doi: 10.3389/fmed.2021.585358 (not cited by the authors, but certainly relevant), and Lythgoe et al 2021, both of which report similarly narrow transmission bottlenecks, at least in pre-VOC viruses.

>>> *We have added the suggested citation.*

It is unfortunately very difficult to say to what extent this common pattern holds true when it comes to the rarer, but epidemiologically relevant, cases of transmission from individuals with longer-lasting and more diverse infections, possibly to other individuals susceptible to multiple infection. The authors make a convincing case for narrow bottlenecks once data are constrained to the type of infection where donor diversity is minimal, but the population-level limitations should be expanded on in the discussion: had there have been any persistent (or severe) infections within the dataset, or any known immunocompromised individuals, would the expectation of narrow bottlenecks still hold? Is diversity of the sampled disease presentations something future studies should specifically address?

>>> *This is indeed an interesting area. We are really unable to speculate on what the bottleneck would be in a persistently infected and/or immunocompromised individual. As such, we're not sure how to discuss. As to the population level implications, the vast majority of transmission events take place among acutely infected individuals such as those studied in our manuscript.*

The paper does not mention severity of presentation, but since these are individuals in community-based studies, my assumption is that most are likely to be cases of relatively mild self-limiting disease. Is it possible to include a description of the range of severity in among the cases, age range, vaccination status if known, treatment, and any notable demographic factors?

>>> *The reviewer is correct that this was a community (household) cohort in which nearly all (if not all cases) had mild self-limiting disease. Therefore, we did not analyze severity or contributors to severity due to limited range in the outcome.*

I have a few minor comments which appear below.

Abstract, line 31 - please specify how many samples were VOC v non-VOC

>>> *This is a good suggestion. However, we are already over the word limit for the abstract and therefore unable capture all the relevant details in Table 1.*

Results line 87 / Figure S2 - can you justify the exclusion on the basis of >2 SNPs? In the cases where putative transmission pairs differed at a small number of consensus positions, as in households HH32 and HH33, was there any evidence of within-host diversity at these positions in the two samples?

>>> *Lythgoe et al. (cited and mentioned by this reviewer) applied this cut off (< 3 consensus differences) in their analysis. We also recently published a study of 26 households in which no pair differed by > 2 consensus differences (was a preprint, now cited Bendall et al. mSphere). In our study, we found that iSNV did not improve transmission inference in households – largely because of tight bottlenecks (they aren't shared). We have found similar results in influenza (see McCrone et al. eLife 2018).*

Results L109 - is "trending" used in the sense of drawing a trend line?

>>> *We changed "trending" to "plotting" for clarity.*

Fig S4 legend - please state what the asterisks indicate.

>>> Thank you for noticing this! We revised the figure legend to clarify that the “asterisks” are actual a triangle and inverted triangle overplotted on each other.

Reviewer #3 (Remarks to the Author):

The size of the transmission bottleneck of a viral pathogens determines how viral genetic diversity is transmitted between hosts. This is an important quantity to estimate to understand how natural selection and drift will act across chains of transmission and ultimately at the host population scale. This paper uses epidemiologically confirmed transmission pairs to estimate the size of the SARS-CoV-2 bottleneck. While similar studies are often constrained by the size of the available dataset the present study has data from 134 putative transmission pairs. Therefore, the authors are able to ask if viral evolution (i.e. the emergence of VOCs) has led to changes in the bottleneck size over time. This is the first study that I am aware of to evaluate this question and therefore represents an important contribution to the literature. The manuscript is very clearly written with pleasant data visualizations and the conclusions in the paper are largely supported by the data analysis.

I do, however, have some comments which I would prefer to see addressed prior to publication.

Major Comments

-Throughout the manuscript (lines 32, 77-78, 160-161, 177-178, etc) the authors assert that limited genetic diversity resulted in a small bottleneck. First, it is not clear whether the authors are implying that the low genetic diversity results in a small estimated bottleneck (lines 160-161) or a small actual bottleneck (lines 32, 77-78, 177-178). The distinction is important as it determines whether this is a comment about the limitations of the inference method or a comment on the biological reality. Regardless, however, I do not think I necessarily agree with either claim.

>>> We think it is a limitation inherent to estimating bottlenecks from sequence data (see other comments addressed below). The reviewer is correct that we could be more clear. The following changes have been made.

Line 32 – “These bottlenecks reflect the low diversity at the time of transmission”

Lines 77-78 – “This tight bottleneck estimate was driven by the limited diversity in the donor host at the time of transmission”

Lines 160-161 “Because of this limited genetic diversity, we estimated a tight bottleneck”

Lines 168-170 – “Much of this diversity is not shared between hosts, and therefore, multiple studies in different settings have measured a tight transmission bottleneck for SARS-CoV-2.”

Lines 177-178– We modified this entire paragraph

“Additionally, we demonstrate that increased transmissibility, whether through force of infection or immune escape, doesn’t change the bottleneck size for SARS-CoV-2. Low genetic diversity can constrain transmission bottleneck estimates. If only a single genotype is transmitted, a bottleneck of 1 is inferred. However, multiple virions of a single genotype can found a population. Transmission of multiple genetically identical virions is more likely when there are few iSNV and/or when iSNV are at a low frequency and when bottlenecks are already reasonably narrow (i.e. <10). Regardless of the ability to detect the actual number of founding virions, the biological effect is the same – no genetic diversity is being transmitted from the donor to the recipient. In our comparison of non-VOC and VOC, the short generation time of SARS-CoV-2 does not allow for diversity to accumulate in the donor, much less transmit..”

First, I do not think it is true that low genetic diversity in the donor host should bias the inference method downwards. While it is true (as is noted) that when there are no iSNVs in the donor host it becomes impossible to estimate the bottleneck size, if there are any variants for the inference method to act on, then there should be no inherent bias due to low diversity in the donor host.

>>> We did not state that there was any bias in the inference method, only that one cannot estimate a bottleneck when there are no iSNV in the donor.

The method is more concerned with the similarities in variant frequencies between donor and recipient hosts. For example, the author's own Figure 3 shows that there are actually quite a few variants present in the donor hosts (all of the black dots with x value > 0) but most of them are present at either 0% or 100% in the recipient host, which is why the estimated bottleneck is low. The same quantity of data would provide very different estimated bottleneck sizes if these donor-recipient frequencies were more similar. While I haven't tested this myself and therefore am willing to be proven wrong, I think that if there was only a single iSNV identified in both the donor and recipient host and if it were present at near identical frequencies in each then the beta binomial method should estimate a large bottleneck size.

>>> *We agree, if there were iSNV at somewhat low frequency that were present in both donor and recipient, one would estimate a larger bottleneck. However, diversity in our sampled individuals was low – both the number of iSNV and their frequency – and with one exception, there were no shared iSNV. 51% of individuals had no iSNV above a 2% threshold and the majority of iSNV were <10%. It is therefore a limitation of the inference given the underlying data – one can't estimate a bottleneck other than 1.*

Second, I further do not think low diversity in the donor host would indicate that the actual bottleneck size is small. Particularly, the phrasing on lines 177-178 is confusing to me. The beta binomial method estimates the census bottleneck size. That is, the number of individual virions that seeded the sampled diversity in a recipient host. These individual virions do not necessarily have to be genetically distinct. While it would be impossible to infer, a completely homogeneous donor viral population could have a census bottleneck size > 1 even if its genetic bottleneck (number of genetically distinct virions) is necessarily fixed at 1. This is a common misconception in the literature but is fairly unambiguously defined in Sobel Leonard et al. 2017: "This transfer is characterized by a transmission bottleneck, defined as the size of the founding pathogen population in the recipient host."

>>> *Yes, we agree. As stated by the reviewer "While it would be impossible to infer..." This limitation is inherent to estimating bottlenecks in natural infections from sequence data. We have tried to highlight this important area of agreement in the modified paragraph above, in the revised manuscript.*

Minor comments:

-I'm having trouble understanding how it was concluded that in nearly all cases the index cases were sampled at peak viral shedding (Line 110-111) from the analysis of only two data points per person, especially given findings that infectiousness peaks right about at symptom onset (<https://doi.org/10.1038/s41591-020-0869-5>), considerably earlier than most of the samples in this analysis. I do not think this has a significant impact on the results or conclusions, however.

>>> *This was indeed overstated in the submitted manuscript. We cannot conclude that they were at the peak. We have revised to state "at or near peak viral shedding."*

-It's not totally clear which sites are masked as the De Maio analysis has evolved over time. I was able to find it by digging through the GitHub repo but I'd like to see the "ncov_references/problematic_sites_v7.txt" file included as a supplementary table to aid in tracking that information down.

>>> *This has been provided as a new Table S4*

-This is not something that I think needs but be done, but I would like to see some visualization of the Nb estimates or to see them in a main table purely for readability. It can be frustrating as a reader to open a paper looking for a very specific estimate (in this case the Nb across clades) and not see it presented in any figures or tables in the main text.

>>> *This is a good suggestion and we have considered it carefully. The summary clade-level Nb have exceedingly small confidence intervals (e.g. 1; 95% CI 1-1 and 2; 95% CI 2-2). Therefore a plot would just have a series of dots at 1 with no appreciable CI brackets and then one dot at 2. We don't think this would be a compelling visualization. The individual level Nb have CI, but there are quite a lot of them to include in the main text (see Table S3). As the reviewer doesn't think this "needs to be done," we have not addressed this comment.*

-I'll leave this to the discretion of the editor but I prefer to see code moved to a DOI'd repository prior to publication. GitHub repos can be deleted and thus are not a permanent storage solution.

>>> *We have supplied this through Zenodo and added the doi to the methods.*

Overall I think this is a wonderful analysis that robustly addresses an important question in the field.

REVIEWERS' COMMENTS

Reviewer #1 (Remarks to the Author):

All my comments have been addressed.

Reviewer #2 (Remarks to the Author):

I would like to thank the authors for answering some of my queries. There is one point outstanding:

>>> The reviewer is correct that this was a community (household) cohort in which nearly all (if not all cases) had mild self-limiting disease. Therefore, we did not analyze severity or contributors to severity due to limited range in the outcome.

This is an important point - please include in the text the age distribution of the participants in the transmission pairs, and any severe cases that did occur. If no demographic data are available, it is best to state this explicitly.

Reviewer #3 (Remarks to the Author):

I thank the authors for their careful consideration of my comments to the original manuscript. I have no remaining major comments, however I do have a few minor points:

1)Figure S2: Is it possible to add an x-axis for the position of the annotated mutations. This plot is very difficult to interpret without this.

2) Figure S3: Is it possible to add x and y axis labels to this plot?

3) Figure S4: The uploaded Figure S4 (a continuation of Figure S3?) is never referenced in the text. The figure S4 referenced on lines 146 and 150 appear to be intended for Figure S5.

4) Line 228/299: Maybe change "typical transmission chain" to "acute transmission chain" to be a bit more precise about the presumed intended meaning of this sentence.

5) Table S4 is not referenced in the text (presumably should be on line 286).

Response to Reviewer Comments

Reviewer #1 (Remarks to the Author):

All my comments have been addressed.

> no additional edits

Reviewer #2 (Remarks to the Author):

I would like to thank the authors for answering some of my queries. There is one point outstanding:

>>> The reviewer is correct that this was a community (household) cohort in which nearly all (if not all cases) had mild self-limiting disease. Therefore, we did not analyze severity or contributors to severity due to limited range in the outcome.

This is an important point - please include in the text the age distribution of the participants in the transmission pairs, and any severe cases that did occur. If no demographic data are available, it is best to state this explicitly.

> age ranges for individuals in both household cohorts are provided in line 218)

Reviewer #3 (Remarks to the Author):

I thank the authors for their careful consideration of my comments to the original manuscript. I have no remaining major comments, however I do have a few minor points:

1)Figure S2: Is it possible to add an x-axis for the position of the annotated mutations. This plot is very difficult to interpret without this.

> we added a scale/legend bar with approximate genome positions to panel B

2) Figure S3: Is it possible to add x and y axis labels to this plot?

> added

3) Figure S4: The uploaded Figure S4 (a continuation of Figure S3?) is never referenced in the text. The figure S4 referenced on lines 146 and 150 appear to be intended for Figure S5.

> corrected this in the text and changed to Nature Communications style (Supplementary Figure 3 and 4).

4) Line 228/299: Maybe change "typical transmission chain" to "acute transmission chain" to be a bit more precise about the presumed intended meaning of this sentence.

> *we changed this to "transmission chains of acutely infected individuals"*

5) Table S4 is not referenced in the text (presumably should be on line 286).

> *corrected and changed to Nature Communications style (Supplementary Data Table 4)*